# Evaluating the Effectiveness of a Novel Systematic Screening Approach for Tuberculosis among Individuals Suspected or Recovered from COVID-19: Experiences from Niger and Guinea

**DOI:** 10.3390/tropicalmed7090228

**Published:** 2022-09-05

**Authors:** Aboubacar Sidiki Magassouba, Souleymane Mahamadou Bassirou, Almamy Amara Touré, Boubacar Djelo Diallo, Soumana Alphazazi, Diao Cissé, Mohamed Sitan Keita, Elhadj Saidou Seyabatou, Adama Marie Bangoura, Hugues Asken Traoré, Tom Decroo, Jonathon R. Campbell, Vanessa Veronese, Corinne Simone Collette Merle

**Affiliations:** 1Department of Public Health, Faculty of Health Sciences and Techniques, Gamal Abdel Nasser University, Conakry BP 1147, Guinea; 2National TB Programme, Conakry BP 1147, Guinea; 3National TB Programme, Niamey 12913, Niger; 4Department of Public Health, Kofi Annan University of Guinea, Conakry BP 1367, Guinea; 5National Centre of Training and Recherche in Rural Health of Mafèrinyah, Forécariah BP 2649, Guinea; 6The Special Programme for Research and Training in Tropical Diseases (TDR), World Health Organization, 1211 Geneva, Switzerland; 7Department of Clinical Sciences, Institute of Tropical Medicine, 2000 Antwerpen, Belgium; 8Department of Epidemiology, Biostatistics and Occupational Health and The McGill International TB Centre, McGill University, 5252 Blvd de Maisonneuve West, Montreal, QC H4A 3S5, Canada

**Keywords:** COVID-19, tuberculosis, active TB screening, Guinea, Niger

## Abstract

Evidence suggests that the COVID-19 pandemic negatively impacts tuberculosis (TB) activities. As TB and COVID-19 have similar symptoms, we assessed the effectiveness of integrated TB/COVID-19 screening in Guinea and Niger. From May to December 2020, TB screening was offered to symptomatic patients after a negative COVID-19 PCR test or after recovery from COVID-19 in Guinea. From December 2020 to March 2021, all presumptive COVID-19 patients with respiratory symptoms were tested simultaneously for COVID-19 and TB in Niger. We assessed the TB detection yield and used micro-costing to estimate the costs associated with both screening algorithms. A total of 863 individuals (758 in Guinea, and 105 in Niger), who were mostly male (60%) and with a median age of 34 (IQR: 26–45), were screened for TB. Reported symptoms were cough ≥2 weeks (49%), fever (45%), and weight loss (30%). Overall, 61 patients (7%) tested positive for COVID-19 (13 in Guinea, 48 in Niger) and 43 (4.9%) were diagnosed with TB disease (35 or 4.6% in Guinea, and 8 or 7.6% in Niger). The cost per person initiating TB treatment was USD $367 in Guinea and $566 in Niger. Overall, the yield of both approaches was high, and the cost was modest. Optimizing integrated COVID-19/TB screening may support maintaining TB detection during the ongoing pandemic.

## 1. Introduction

Globally, an estimated 10 million people fell ill with tuberculosis (TB) in 2020, with an estimated 1.5 million TB deaths [1]. TB was the leading cause of death from a single infectious agent until the arrival of a newly emerging novel pathogen (SARS-CoV-2) which caused the 2019 coronavirus (COVID-19) pandemic and has since become the leading cause of death from an infectious disease [2]. The COVID-19 pandemic quickly created a global threat to public health with devastating consequences regarding the availability of resources for healthcare systems and providers and wide-reaching effects on social and economic stability [3]. This impact has been felt by the TB response, with studies suggesting that the COVID-19 pandemic has led to reductions in TB screening [4,5] case detection [6,7], and an increase in TB-related mortality [7,8].

As both diseases primarily affect the lungs, people with COVID-19 and/or TB disease share similar symptoms, including cough, fever, and difficulty breathing [9]. Given this similarity, the differential diagnosis between the two conditions remains a challenge. Both conditions also share certain determinants of mortality, namely age and poverty, as well as co-morbidities such as HIV coinfection [10]. These findings highlight the importance of reinforcing TB prevention and care activities to avoid significant disruption to TB control efforts, as well as TB-related morbidity and mortality caused by the COVID-19 pandemic.

Guidance on how to ensure the continuity of essential services for people affected by TB during the COVID-19 pandemic has been developed by the World Health Organization (WHO), which promotes innovative person-centered approaches and integrated interventions to fight the two diseases [9]. Guinea and Niger are two low-income countries facing the COVID–19 pandemic, which occurs against a backdrop of an annual TB incidence of 176 and 84 cases per 100,000 persons, respectively [1]. The majority of TB case finding in each country is passive, though some active case finding activities occur. The experience of Ebola virus disease in Guinea has shown the degree of destabilization that uncontrolled epidemics can have on the health system, including TB prevention efforts, as well as broader socioeconomic consequences, and the attendant impacts on health [11,12]. In Guinea and Niger, the COVID-19 pandemic has exacerbated current challenges facing national TB programs (NTPs), affecting case finding and underdiagnosis and/or underreporting of tuberculosis cases. Despite efforts, TB treatment coverage in Guinea and Niger was 66% and 56%, respectively, in 2020 [13].

The symptomatic similarities between the two conditions provide an opportunity for integrated screening approaches, which may increase opportunities for detection and diagnosis and maximize the judicious use of resources. This study describes an evaluation of the effectiveness and costs of novel strategies to enhance TB screening among individuals with suspected or confirmed COVID-19 to mitigate the impact of the COVID-19 pandemic on TB prevention and care efforts in Guinea and Niger.

## 2. Materials and Methods

### 2.1. Study Design, Period and Population

A cross-sectional, multi-center study was conducted between May and December 2020 in Guinea, and between December 2020 and March 2021 in Niger, to evaluate the effectiveness and costs of an enhanced TB screening strategy in the context of the COVID-19 pandemic. In Guinea, eligible individuals were defined as those who (1) tested negative for COVID-19 via polymerase chain reaction (PCR), either following suspicion of COVID-19, or recovery from a previously confirmed (by PCR test) case of COVID-19; (2) self-reported at least one of the following signs of TB: cough, fever, night sweats and/or weight loss, and; (3) agreed to participate in the research.

In Niger, eligible individuals were defined as those (1) in whom COVID-19 was suspected, either on the basis of signs or symptoms of COVID-19 and/or self-presentation to a COVID-19 testing facility, and regardless of PCR result; (2) who self-reported one or more of the following signs of TB: cough lasting more than two weeks, fever, night sweats and/or weight loss, and; (3) agreed to participate in the research (Table 1).

### 2.2. Setting

The study was conducted in Conakry and Niamey, the capital cities of Guinea and Niger, respectively, which have recorded the highest number of COVID-19 cases in each country and were the locations of the first identified cases [14]. In Conakry and Niamey, there were five and four COVID-19 screening and treatment centers, respectively.

All COVID-19 centers in both countries offer free screening for COVID-19 using PCR tests for residents with a delay of between 24 and 72 h, depending on the volume of testing. In both countries, a list of close contacts is established after the discovery of a confirmed COVID-19 case, and they are then monitored by health care staff for 14 days for the development of signs and/or symptoms suggestive of COVID-19 and referral for COVID-19 testing as required. 

### 2.3. Intervention/Strategy

The broad aim of the intervention was to integrate TB and COVID-19 screening activities to enhance TB detection in the context of the COVID-19 pandemic in Guinea and Niger. The intervention was implemented in concert with the structures that oversee the surveillance of the COVID-19 pandemic in Guinea (the Health Security Agency) and in Niger (the Ministry of Health).

In each country, a list of persons meeting the respective eligibility criteria (Table 1) was compiled by the research team in collaboration with the institutions in charge of COVID-19. Health care workers were responsible for contacting these patients and monitoring the presence of at least one of the following symptoms of TB: cough lasting two or more weeks, fever, night sweats, and/or weight loss (defined as an unintentional decrease in body weight). Anyone who reported at least one of these signs was invited for TB screening.

TB screening of eligible individuals was managed by trained community health workers who either referred individuals to the nearest TB diagnosis and treatment centre for sputum collection and testing or collected sputum from the individual at their homes if necessary. All samples were tested using either smear microscopy or Xpert MTB/RIF testing, while samples collected at home were transported on motorbikes using coolers to the nearest laboratory (Figure 1). 

### 2.4. Study Variables 

Data on the following variables were collected from participants and analyzed during this study: socio-demographic characteristics (country of origin, age in years, sex, current occupation, residence), presence of TB-related symptoms (yes/no: cough lasting more than two weeks, fever, weight loss, night sweats, other signs), recent contact (defined as within the last 6 months) with a TB case (yes/no), history of anti-TB treatment (new or previous treatment unknown, previously treated or relapse), bacteriologically confirmed diagnosis of TB (yes/no) with microscopy or Xpert MTB/RIF result, and COVID-19 test results (PCR positive, PCR negative). The operational definitions used for TB and COVID-19 are those published by the WHO [9,10].

To evaluate the costs of the enhanced screening strategy, we did micro-costing (a bottom-up approach) to estimate the costs associated with each integrated screening approach in each country. We considered the unit costs of healthcare worker training, symptom screening, specimen transport, testing with sputum smear microscopy and Xpert MTB/RIF, and TB disease treatment. We did not include costs associated with research nor COVID-19 testing (since all patients would be tested for COVID-19 without our intervention). Costs were from the healthcare system perspective in 2020 USD ($)—we used purchasing power parity and exchange rates to convert costs to USD as appropriate. To estimate personnel costs, we used time estimation questionnaires to determine the amount of time that was required for each task and used personnel salaries to arrive at a cost value. We used the Global Drug Facility to estimate the costs of the materials required for smear microscopy and Xpert MTB/RIF to estimate the costs of the medications included in TB disease treatment (6 months of isoniazid and rifampin, supplemented with ethambutol and pyrazinamide during the first 2 months). Capital costs of equipment (for smear microscopy or Xpert MTB/RIF) were annuitized over a five-year time horizon. Training costs were derived from the actual realized costs of conducting training and were prorated, assuming training would need to be repeated annually. Overhead costs associated with the laboratory were derived from previous estimations in Benin [15], which used manpower allocation and expense logs to determine the overhead costs associated with Xpert MTB/RIF and smear microscopy.

### 2.5. Sampling Procedure and Data Collection

We consecutively recruited all eligible individuals who attended the study sites during the study period. Data were captured by research team members using an ODK-based form accessed via android phones. Research team members were trained in advance on how to enter data and use ODK and android phones. Patients were uniquely identified, and data quality control was performed periodically through routine supervision and analysis by senior research staff members.

### 2.6. Data Analysis

Frequencies (percent) or means (standard deviation) were used to describe categorical and continuous variables, respectively. Significance tests were performed using the Chi-square test (or exact Fischer’s test where appropriate) for categorical variables and the Student’s *t*-test for quantitative variables to compare differences in proportions reported by Niger and Guinea.

Using the cost data collected and the number of evaluations performed during the study (e.g., number of symptom screens, smear microscopy performed), we calculated the overall cost associated with the integrated screening approach. From this value, we estimated the cost per patient screened for TB symptoms (by dividing the overall cost of the approach by the number of patients screened) and the cost per patient initiating TB disease treatment in each country (by dividing the overall cost of the approach by the number of patients initiating TB treatment).

### 2.7. Ethical Considerations

The study was approved by the National Ethics Committees of Guinea (CNERS) and Niger (CNERS) according to numbers 94/CNERS/20 and 63/2020/CNERS, respectively. No personally identifiable information was collected, and the data were stored on a server that only authorized persons had access to. Informed consent was obtained from patients before enrolling them in the study.

## 3. Results

During the response period, 2826 people were tested for COVID-19 in Guinea and Niger, with results known for 2766 (97.8%). Of these, 863 (30.5%) reported at least one sign of TB and were subsequently tested, of whom 758 (26.8%) were in Guinea and 105 (3.7%) in Niger. TB was found in 43 patients, 5 of whom were co-infected with TB and COVID-19 (Figure 2).

Overall, most participants who were tested for TB were male (60.4%), aged 15–30 years (38.5%), and reported no history of TB (94.6%). Cough lasting more than two weeks, fever, and weight loss were reported by 49.6%, 44.6%, and 30.2%, respectively. Most participants did not report recent contact with a TB case (83.2%). Significant differences were found between participants from Guinea and Niger regarding sex, age, and the proportion of participants reporting weight loss and recent contact with a TB case (Table 2). Among those tested for TB, a total of 61 patients (7.1%) were PCR-positive for COVID-19 (13 in Guinea (1.7%) and 48 in Niger (46%)). In total, TB was diagnosed in 43 participants (5%) with a slightly higher proportion of TB cases identified in Niger (8/105; 7.6%) than Guinea (35/758; 4.6%; *p* = 0.2). Among those with TB, TB was bacteriologically confirmed in 37/43 (86%) participants and clinically diagnosed in 6/43 (14%) participants. In Niger, where, unlike Guinea, the target population included individuals with confirmed COVID-19, five of the eight (63%) patients diagnosed with TB disease also had COVID-19 (Table 2). Of these five patients, one died before initiating TB treatment and the other four commenced TB treatment with good outcomes (data not reported).

### Cost Assessment

Costs are shown in Table 3 for the population with known COVID-19 test results (*n* = 2766). We estimated that the overall cost of the integrated screening program for 2,640 patients was $12,840 in Guinea, equivalent to $4.87 per patient screened for TB symptoms. In Niger, the overall cost for 126 patients was $3960, equivalent to $31.43 per patient screened for TB symptoms. The cost per patient initiating TB disease treatment was $367 in Guinea and $566 in Niger. Costs in Niger were largely driven by comparatively higher costs associated with TB symptom screening such as capital costs associated with the technology required to link patients with symptoms for further follow-up.

## 4. Discussion

This study demonstrated that an integrated approach to COVID-19/TB screening in two high-TB burden settings can yield a high number of TB diagnoses while incurring modest costs. The yield of TB diagnoses ranged from 1.5% among all individuals under suspicion of COVID-19 to 5% when restricted only to those patients who also had symptoms suggestive of TB. The yield found in this study is extremely high and well above WHO’s stated thresholds for systematic TB testing [16]. In addition, in Niger, 63% of patients diagnosed with TB also had COVID-19. The costs associated with screening each patient were modest and could be reduced with higher patient volumes, thus reducing fixed costs.

Despite recommendations from global health bodies [16,17,18], there are limited data available on the impact of integrated COVID-19/TB screening approaches. While there are some examples of COVID-19 screening integrated with TB screening activities [19,20], to our knowledge, this is the first study to assess the costs and effectiveness of integrating routine TB screening among individuals suspected of COVID-19, or those recovered from COVID-19 with persisting TB symptoms. In Guinea, TB was only screened for among those without COVID-19 (negative or recovered), while in Niger, TB was screened for among all individuals, regardless of COVID-19 status. However, in both countries, the TB screening approach was based on the existence of TB symptoms. The approaches used by the two countries considered the disease incidence and testing capacity for COVID-19 as the only laboratory diagnostic approach approved in these two countries is PCR for the detection of SARS-CoV-2, as this test is considered the most accurate. However, the centres with technologies and qualified human resources for performing PCR testing for COVID-19 test are mainly located in the cities of Conakry and Niamey, which poses logistical challenges for the inclusion of suspected COVID-19 patients from other cities and limits the potential for a wider scale-up of the integrated screening approach. The use of alternative testing technologies that have both high sensitivity among symptomatic COVID-19 patients and wider availability at peripheral sites should be considered. Multiplexing NAAT platforms such as GeneXpert have been recommended for simultaneous, integrated testing of COVID-19 and TB [17]. The use of rapid antigen tests, which are inexpensive and widely available, has also been suggested in contexts where molecular testing is limited or unavailable or when long turnaround times hamper patient triaging and response [18] and could facilitate the scale-up of integrated screening in areas with insufficient laboratory access in Guinea and Niger.

In our study, all the individuals considered eligible who were invited for screening agreed to participate (data not reported), suggesting a very high level of acceptability for combined TB/COVID-19 screening among individuals in Guinea and Niger. The high participation was achieved with the help of health care workers and community agents who sensitized patients to screening while explaining the similarity of the symptoms of these two diseases. This acceptability can be aligned with the high acceptance group for TB screening [21].

The typical triad of symptoms presenting in the majority of TB patients consists of fever, night sweats, and weight loss, while a persistent non-remitting cough is the most frequently reported symptom [22,23,24,25]. In our study, we used this symptom screen to identify the need for further diagnostic testing for TB. However, to increase the potential yield of integrated algorithms, chest radiography could be considered in addition to symptom screening in diagnostic algorithms for TB. The use of computer assisted detection (CAD) software for the interpretation of digital chest X-rays (CXR) has recently been recommended in both TB screening and triage contexts among individuals aged 15 years and above by WHO as an alternative to human readers [16]. CXR is a highly sensitive screening tool and digital reading of CXR by CAD software can overcome barriers associated with limited trained readers, ensure appropriate allocation of costly diagnostic testing such as GeneXpert, and can help identify asymptomatic or pre-symptomatic cases otherwise missed by symptom screening alone [26]. The integration of CAD has demonstrated the potential for increasing the reach and yield of integrated screening approaches for COVID-19/TB [19] and could similarly be considered in the West African context.

Our integrated screening approach has been successful, with an overall TB case detection of 1.5% among those presenting with COVID-19 to testing centres in two high TB burden cities. Depending on the country and the strategy, the proportion of participants with symptoms who were ultimately diagnosed with TB was, respectively, 7.6% and 4.6% in Niger and Guinea. We expect these yields to vary in other settings based on local TB epidemiology. Of the patients diagnosed with TB in Niger, five were co-infected with COVID-19. This is important, especially as it results from active screening in patients who would normally be missed by the routine TB screening system put in place by the NTP. Current evidence suggests that while people with TB are not necessarily more likely to contract COVID-19, those with COVID-19 are more likely to develop severe complications when coinfected with TB [24]. The impact of sequelae among patients with TB/COVID-19 coinfection, and the need for further rehabilitation of patients with sequelae, requires further evaluation [19].

In this study, we looked for TB in presumptive and recovered after COVID-19 patients which required the significant coordination of all actors involved and required a synergy of action from all actors involved in the fight against TB and COVID-19, as well as a dynamic system of transporting sputum samples through the rational use of community workers. A reverse approach of testing patients with TB for COVID-19 would also be interesting to improve the diagnosis and management of the two diseases yet would require an additional testing capacity given the increased testing demands, and its overall efficiency would depend on current epidemiology. Despite this, there are opportunities for implementing this approach, especially in low-income countries, where the new Xpert^®^Xpress SARS-CoV-2 (Cepheid, Sunnyvale, CA, USA)diagnostic [22,23] on the GeneXpert platform can be used to diagnose COVID-19.

### Difficulties and Limitations of the Study

The overall effectiveness of this strategy may have been constrained by some of the real-world challenges such as the lack of timely bacteriological (in particular microscopy, which takes approximately 24 to 72 h), and COVID-19 test results, the decreasing number of suspected cases of COVID-19 at times, and a sense of fear among laboratory workers becoming infected by COVID-19 which occasionally created laboratory bottlenecks. Additionally, our study was conducted in locations selectively chosen based on their diagnostic capacity; how such an approach would work with tests not as reliant on laboratory services should be evaluated. Though the diagnostic algorithms in Niger and Guinea were similar, they had important differences. The approach in Niger took advantage of a “one-stop-shop” type approach where everyone with symptoms would receive a TB and COVID-19 test and TB/COVID-19 coinfection would be detected simultaneously. However, the approach in Guinea would miss TB/COVID-19 coinfections (as only those negative or already recovered from COVID-19 were screened for TB) and delay the initiation of TB disease treatment. Due to the relatively smaller sample size in Niger, however, we could not perform statistical comparisons between sites; moving forward, approaches will need to be adapted to local contexts. Finally, multiple means of TB diagnosis (sputum smear microscopy and Xpert MTB/RIF) were used, and chest radiography was not incorporated into algorithms. Differing sensitivities of smear microscopy and Xpert may have resulted in missed diagnoses of TB.

## 5. Conclusions

Systematic screening for TB in suspected or recovered after COVID-19 patients presenting with persistent respiratory signs is a feasible strategy with modest cost. In our settings, the yield of bidirectional TB and COVID-19 screening among persons presenting to COVID-19 testing centers was high, suggesting an efficient method for improving case finding among this symptomatic population, which remain an important reservoir.

## Figures and Tables

**Figure 1 tropicalmed-07-00228-f001:**
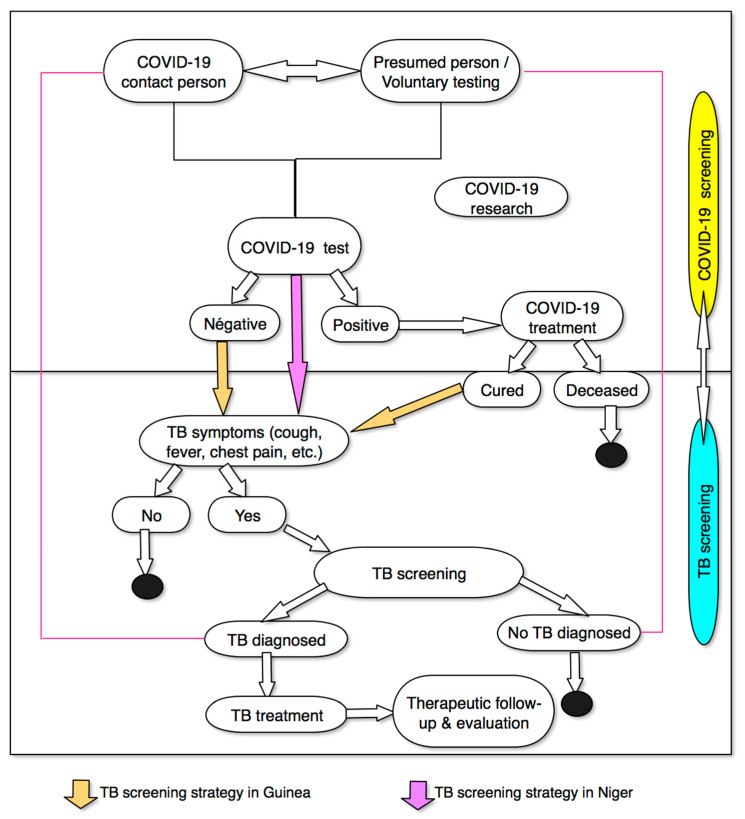
Integrated TB/COVID-19 screening algorithm used in the study.

**Figure 2 tropicalmed-07-00228-f002:**
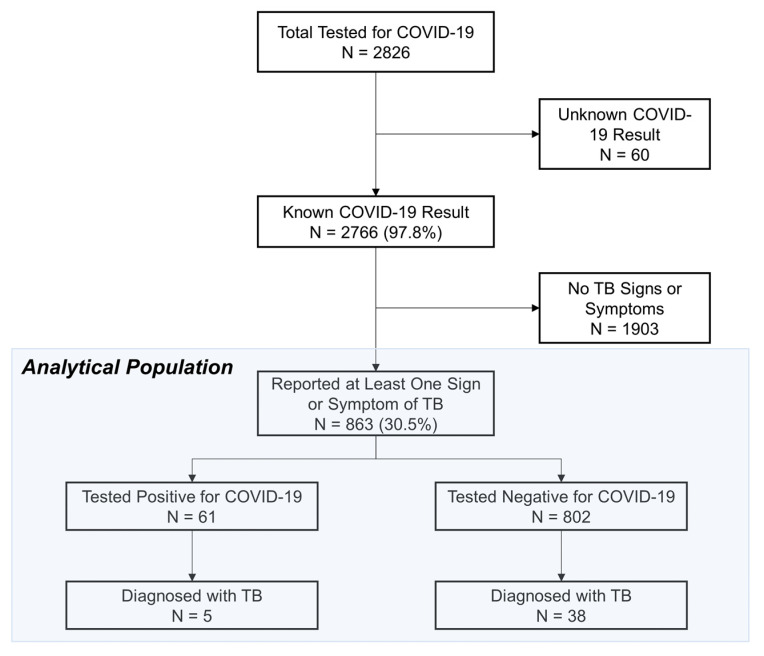
Study flow diagram.

**Table 1 tropicalmed-07-00228-t001:** Integration of TB screening in COVID-19 care in Niger and Guinea.

	Guinea	Niger
**Design**	Cross-sectional study	
**Period**	May to December 2020	December 2020 to March 2021
**Setting**	5 COVID-19 screening and treatment centers in Conakry	4 COVID-19 screening and treatment centers in Niamey
**Study population**	Persons with a negative COVID-19 test or recovered from COVID-19 disease and with symptoms of TB	Persons with presumptive COVID-19 and with symptoms of TB regardless of the COVID-19 test result
**COVID-19 diagnostic method**	Polymerase chain reaction (PCR) test	
**TB diagnostic method**	Sputum smear microscopy (SSM) and/or Xpert MTB/RIF	
**Sample transportation and data collection**	Health workers/community workers	

**Table 2 tropicalmed-07-00228-t002:** Socio-demographic and clinical characteristics of individuals screened for TB/COVID-19 in Guinea and Niger.

	Total	Guinea	Niger	
	*n* (%)	*n* (%)	*n* (%)	*p*-Value ^#^
**Total**	863	758	105	
**Gender**				0.004
Female	342 (39.6)	314 (41.4)	28 (26.7)	
Male	521 (60.4)	444 (58.6)	77 (73.3)	
**Age group**				<0.001
<=15	27 (3.1)	26 (3.4)	1 (1.0)	
>15–30	332 (38.5)	311 (41.0)	21 (20.0)	
>30–45	289 (33.5)	261 (34.4)	28 (26.7)	
>45–50	48 (5.6)	40 (5.3)	8 (7.6)	
>50–65	104 (12.1)	89 (11.7)	15 (14.3)	
>65	63 (7.3)	31 (4.1)	32 (30.5)	
**History of TB**				0.4
No	816 (94.6)	714 (94.2)	102 (97.1)	
Yes	40 (4.6)	38 (5.0)	2 (1.9)	
Unknown	7 (0.8)	6 (0.8)	1 (1.0)	
**Cough (≥2 weeks)**				0.1
No	435 (50.4)	389 (51.3)	46 (43.8)	
Yes	428 (49.6)	369 (48.7)	59 (56.2)	
**Fever**				0.7
No	473 (54.8)	419 (55.3)	54 (51.4)	
Yes	385 (44.6)	335 (44.2)	50 (47.6)	
Unknown	5 (0.6)	4 (0.5)	1 (1.0)	
**Weight loss**				<0.001
No	599 (69.4)	545 (71.9)	54 (51.4)	
Yes	261 (30.2)	211 (27.8)	50 (47.6)	
Unknown	3 (0.3)	2 (0.3)	1 (1.0)	
**Recent TB contact**				0.02
No	718 (83.2)	621 (81.9)	97 (92.4)	
Yes	140 (16.2)	133 (17.5)	7 (6.7)	
Unknown	5 (0.6)	4 (0.5)	1 (1.0)	
**COVID-19 Test Result**				
Negative	802 (92.9)	745 (98.3)	57 (54.3)	
Positive	61 (7.1)	13 (1.7)	48 (45.7)	
**Final TB Diagnosis**				0.2
No TB	820 (95.0)	723 (95.4)	97 (92.4)	
TB	43 (5.0)	35 (4.6)	8 (7.6)	
Diagnosed with Xpert MTB/RIF Alone	14 (1.6)	11 (1.5)	3 (2.9)	
Diagnosed with SSM Alone	20 (2.3)	17 (2.2)	3 (2.9)	
Both Xpert & SSM Positive	3 (0.3)	2 (0.3)	1 (1.0)	
Clinically Diagnosed *	6 (0.7)	5 (0.7)	1 (1.0)	
**Xpert MTB/RIF**				<0.001
Negative	59 (6.8)	21 (2.8)	38 (36.2)	
MTB detected	17 (2.0)	12 (1.6)	5 (4.8)	
Xpert not done	787 (91.2)	725 (95.6)	62 (59.0)	
**Sputum smear microscopy (SSM)**				
SSM negative *	772 (89.5)	701 (92.5)	71 (67.6)	
SSM positive	27 (3.1)	22 (2.9)	5 (4.8)	
SSM not done	64 (7.4)	35 (4.6)	29 (27.6)	

^#^ As estimated by Chi-square test (or exact Fischer’s test where appropriate) for categorical variables and the Student’s *t*-test for continuous variables between Niger and Guinea. * Clinical diagnoses were made on the basis of symptoms and/or radiological evaluations.

**Table 3 tropicalmed-07-00228-t003:** Cost assessment of TB screening among presumed and/or recovered individuals in Guinea and Niger.

Cost Parameter	Guinea			Niger		
	Number of Patients	Cost per Patient (2020 USD)	Total Cost (2020 USD)	Number of Patients	Cost per Patient (2020 USD)	Total Cost (2020 USD)
Training of Health Workers	--	--	$641	--	--	$143
Symptom Screening	2640	$2.70	$7128	126	$14.70	$1852
Smear Microscopy	723	$1.88	$1359	76	$2.56	$195
Xpert MTB/RIF	33	$16.28	$537	43	$23.33	$1003
TB disease treatment	35	$90.97	$3184	7	$109.62	$767

**Overall cost**	**$12,849**			**$3960**		
Cost per patient screened for TB disease	$4.87			$31.43		
Cost per patient initiating TB disease treatment	$367			$566		

## Data Availability

Requests for original datasets used in this manuscript can be directed to the corresponding author.

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
