# Peer review of "Evaluating the Effectiveness of a Novel Systematic Screening Approach for Tuberculosis among Individuals Suspected or Recovered from COVID-19: Experiences from Niger and Guinea"

_tropicalmed, 2022, doi:10.3390/tropicalmed7090228_

Round 1
Reviewer 1 Report
This manuscript reports on integrated services for tuberculosis (TB) and COVID-19 case finding in Guinea and Niger in order to mitigate against declining TB notifications due to the adverse impact the COVID-19 pandemic has had (and may continue to have) on TB services. Estimated cost per TB case detected is also documented.
Suggestions for major revision:
* For readers to place active TB case finding in COVID-19 services into the overall context, it would be good to include a brief explanation about how TB detection in Guinea and Niger was conducted before the pandemic (and continued during the pandemic?), in other words, are all persons (or persons with respiratory symptoms) attending health services offered symptom-based screening for TB, is there targeted mass TB screening among key populations, etc.
* Cost assessment: I cannot find any explanation on how unit costs per patient were arrived at. This may deserve some further information. There may be some calculations errors in Table 3 (for example, 2,640 x $2,70 = $7,128 rather than 7,119).
* I agree with your interpretation of a high yield of TB among these patients. Is this finding suggestive of health services missing (a large number?) people with TB? If so, what lessons should be drawn and what recommendations should be made?
* You also recognise that had more persons been tested with rapid molecular tests, the yield of TB case finding could have resulted in a higher number of individuals with TB (line 310). This point needs to be expanded - why so few persons were offered Xpert, especially in Guinea, what are the plans for offering rapid molecular tests for TB for all people with presumed TB, etc.
Suggestions for minor revision:
* Line 72: provide (in plural).
* Line 114, 182: you appear to be listing symptoms rather than signs. Same comment for Figure 1 (in the middle below the line separating TB screening from COVID-19 screening).
* Figure 1: no TB diagnosed would read better than TB undiagnosed.
* Line 229: please consider rephasing, for example 'Persons with .... were screened for TB...'.
* Line 282, 313: you have used the word 'cured' in relation to COVID-19 - it may be better to rephrase and refer to recovery after COVID-19.
* Line 316: case finding reads better than case certainment (in English).
Author Response
Response to Reviewer 1 Comments
* For readers to place active TB case finding in COVID-19 services into the overall context, it would be good to include a brief explanation about how TB detection in Guinea and Niger was conducted before the pandemic (and continued during the pandemic?); in other words, are all persons (or persons with respiratory symptoms) attending health services offered symptom-based screening for TB, is there targeted mass TB screening among key populations, etc.
Point 1. * Cost assessment: I cannot find any explanation on how unit costs per patient were arrived at. This may deserve some further information. There may be some calculations errors in Table 3 (for example, 2,640 x $2,70 = $7,128 rather than 7,119).
Answer n° 1. We applied the ingredients approach (micro-costing or bottom-up approach). We identified component costs of each of the main unit costs, collected these costs, and calculated a unit cost for each cost item. For example, Xpert MTB/RIF considered costs of personnel, the machine, maintenance, the cartridge, and lab overhead. We have included additional text in the Data Analysis section to describe the cost synthesis. In brief, using the unit costs for each activity, we multiplied those costs by the number of evaluations performed (this is contained in Table 3). We then divided the total cost of the intervention by the number of patients screened (to get cost per patient screened) or number of patients treated (to get cost per patient initiating TB treatment). The error in calculation was due to rounding—the total was calculated on the precise value, but the table rounded the cost estimate. However, for consistency, we have recalculated using the reported values in the table and reflected this change in the text. The cost per patient screened becomes 4.87 instead of 4.86.
Point 2. * I agree with your interpretation of a high yield of TB among these patients. Is this finding suggestive of health services missing (a large number?) people with TB? If so, what lessons should be drawn and what recommendations should be made?
Answer n° 2. We agree that this may be suggestive of missing people with TB. In fact, this was one of the reasons for this study. In the introduction, we highlight that TB treatment coverage is 66% in Guinea and 56% in Niger in 2020, suggesting many people with TB are missed. Therefore, one of the goals of the study was to help identify these missed people with TB using COVID-19 testing centres. Symptom screening for TB is inexpensive, and so too are rapid antigen tests for covid-19 screening, and we have suggested they be leveraged in dual screening programs (line 243). In addition, we noted that screening could be performed in the reverse direction to search for covid-19 cases in patients with suspected/confirmed TB (line 285).
Point 3. * You also recognise that had more persons been tested with rapid molecular tests, the yield of TB case finding could have resulted in a higher number of individuals with TB (line 310). This point needs to be expanded - why so few persons were offered Xpert, especially in Guinea, what are the plans for offering rapid molecular tests for TB for all people with presumed TB, etc.
Answer n° 3. The Xpert MTB RIF tests were performed based on the availability of the test according to the current algorithms of the two programs because microscopy is still the dominant form of TB diagnosis. Some sites involved in the study do not have routine access to GeneXpert devices; in some cases, we transport patient samples to the GeneXpert sites. Certain presumed cases are nevertheless systematically the subject of the GeneXpert test (case of retreatment, PLHIV, child, etc.). According to the NTPs, there are plans to expand access to Xpert to address these diagnostic bottlenecks.
Suggestions for minor revision:
Point 4. * Line 72: provide (in plural).
Answer n° 4. Ok! Corrected.
Point 5. * Line 114, 182: you appear to be listing symptoms rather than signs. Same comment for Figure 1 (in the middle below the line separating TB screening from COVID-19 screening).
Answer n° 5. Ok! Corrected.
Point 6. * Figure 1: no TB diagnosed would read better than TB undiagnosed.
Answer n° 6. We have also taken this comment into account.
Point 7. * Line 229: please consider rephasing, for example, 'Persons with .... were screened for TB...'. We consider this question to have already been answered, or perhaps it is an error in the number. Please clarify the question if it is not answered.
Answer n° 7. We have also taken this comment into account.
Point 8. * Line 282, 313: you have used the word 'cured' in relation to COVID-19 - it may be better to rephrase and refer to recovery after COVID-19.
Answer n° 8. We have also taken this comment into account.
Point 9. * Line 316: case finding reads better than case certainment (in English).
Answer n° 9. We have also taken this comment into account.

Reviewer 2 Report
The paper "Evaluating the Effectiveness of a Novel Systematic Screening Approach for Tuberculosis among Individuals Suspected or Recovered from COVID-19: Experiences from Niger and Guinea" is fascinating. My notes are as follows:
1. Please check and change the () on the citation. There have to be [ ].
2. Please illustrate better and state why is the reason for the fact about how an integrated approach to COVID-19/TB screening can yield a high number of TB diagnoses while incurring modest costs.
3. About the study's limitations.... are significant bottlenecks? And how important in terms of costs are they? And how about co-infection vs. costs?
4. Please indicate future directions of the study and explore why and where this intervention can be a good idea to show results like in this study.
Author Response
Response to Reviewer 2 Comments
The paper "Evaluating the Effectiveness of a Novel Systematic Screening Approach for Tuberculosis among Individuals Suspected or Recovered from COVID-19: Experiences from Niger and Guinea" is fascinating. My notes are as follows:
Point 1. Please check and change the () on the citation. There have to be [ ].
Answer n° 1. We have changed all references with brackets.
Point 2. Please illustrate better and state why is the reason for the fact about how an integrated approach to COVID-19/TB screening can yield a high number of TB diagnoses while incurring modest costs.
Answer n° 2. This statement is based on the results we found in the study compared to the WHO threshold for routine TB screening. We have noted this aspect in the discussion in paragraph 214-223.
Point 3. About the study's limitations.... are significant bottlenecks? And how important in terms of costs are they? And how about co-infection vs. costs?
Answer n° 3. Some problems were encountered during the study, such as delays in the delivery of some results (COVID-19 or TB) due to laboratory bottlenecks. As we did not consider costs of COVID-19 testing, as this would normally happen, we can only comment on costs of delays for TB. In general, this would not increase costs, but would might delay diagnosis and potentially transmission from those infected but awaiting a diagnosis. With respect to co-infection vs. costs, we have amended our introductory discussion sentence to reference that these are two relatively high TB burden cities. Cost per TB patient initiating TB disease treatment will be directly related to the prevalence of TB in settings. We would expect higher costs in lower prevalence settings, and lower costs in higher prevalence settings.
Point 4. Please indicate future directions of the study and explore why and where this intervention can be a good idea to show results like in this study
Answer n° 4. Thank you. We have stated that scaling up through the use of rapid antigenic tests for covid-19 that are inexpensive in both countries was suggested (line 243). Similarly, we noted that screening could be done in reverse to find covid-19 in suspected/confirmed TB patients (line 285). In addition, we have commented on considering local TB epidemiology before implementing such programs, as this is the main driver of yield and cost-effectiveness.
